

# Validation of a traceable efficiency determination method for wind turbines with a focus on measurement uncertainty

Nijan Yogal[1], Christian Lehrmann[1], Paula Weidinger[2], Zihang Song[2], Hongkun Zhang[3], Maximilian Zweiffel[4], Christian Mester[5]

[1]Explosion-protected Electrical Drive Systems, Physikalisch-Technische Bundesanstalt, Braunschweig, 38116, Germany
[2]Realisation of Torque, Physikalisch-Technische Bundesanstalt, Braunschweig, 38116, Germany
[3]System Validation Mechanical Drive Train, Fraunhofer Institute for Wind Energy Systems, Bremerhaven, 27572, Germany
[4]Chair for Wind Power Drives, RHWT Aachen University, Aachen, 52074, Germany
[5]Eidgenössisches Institut für Metrologie METAS, Bern-Wabern, 3003, Switzerland

*Correspondence to*: Nijan Yogal (nijan.yogal@ptb.de)

**Abstract.** All around the world, new plans and strategies are being implemented for installing energy-efficient wind turbines in order to increase renewable energy capacities and meet the demand for clean energy. Advances in wind turbine technologies and their extremely sophisticated systems have led to an increasing need for proper efficiency determination methods, both in the field and on test benches, to keep up with the accelerated deployment of wind turbines around the globe. This paper

examines the feasibility and practicability of new wind turbine efficiency determination methods on test benches that are based on international standards for electrical machines. Test methods and the static calibration certificates of the employed specialised instruments and sensors were used to evaluate the efficiency and associated measurement uncertainties in ambient conditions. Furthermore, the results were compared with those obtained through the innovative iso-efficiency method, which comprehensively maps the wide operating range of wind turbines.

## 20  1 Introduction

The efficiency of new wind turbines is a critical factor in determining their viability as a clean and reliable source of electrical energy. As awareness increases of the benefits of wind turbines with higher efficiency and power quality, so does the chance for their adoption in the field and contribution to a more stable national power grid. Energy-efficient wind turbines are essential for expanding renewable energy sources and meeting clean energy needs, making efficiency a key consideration for both wind

park operators and policymakers. This article aims to support test bench operators, political decision-makers, and end users such as wind park operators in precisely determining wind turbine efficiency and losses on test benches. To predict the performance of wind turbines quickly and precisely, it is crucial to apply highly accurate measurement methods that are based on national and international standards and guidelines. These methods must also be traceable to national standards to ensure the highest possible efficiency with the lowest possible measurement uncertainty (MU). This is particularly important given

European plans to derive 32 % of its consumed energy from renewable sources by 2030 ("Renewable energy | Fact Sheets on the European Union," n.d.). Wind energy investors are looking for wind turbines that are more efficient, provide a quicker

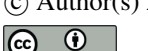



return on investment, and have a longer lifespan. Wind turbines are intricate and sophisticated systems that require intensive bench testing prior to installation in the field, so accurate measurement methods are of utmost importance. But the standardised tests required for determining the efficiency of wind turbines on test benches are not yet available. Generally speaking, any

available standardised method for determining the efficiency of rotating electrical machines could be used to determine the efficiency of wind turbines. Existing methods, however, do not take account of the wind turbine-specific property of operating points as different combinations of torque $T$ and rotational speed $n$, and therefore measure too few relevant operating points. In the EMPIR project WindEFCY ("Traceable mechanical and electrical power measurement for efficiency determination of wind turbines," n.d.), a method for determining the efficiency of wind turbines on test benches was developed based on the

two input variables of torque $T$ and rotational speed $n$. Using this method, a traceable and precise iso-efficiency map can be generated over the entire working range of the wind turbine.

This paper focuses on validating this newly developed iso-efficiency map method for the traceable efficiency determination of wind turbines on test benches. To this end, the efficiency of an asynchronous machine (ASM) was determined along with the corresponding MU on a small-scale test bench (SSTB) using both the standardised methods and the newly developed

efficiency map method, as outlined in Table 1. In addition, our prior research papers ((Yogal et al., 2021) and (Weidinger et al., 2021b)) encompassed a thorough literature review of efficiency measurement methods and relevant national and international standards.

**Table 1: Summary of standards to determine the efficiency of ASMs and provide information on uncertainty.**

| Measurement method | International standard | Total load points | Complexity to perform | Remarks on measurement uncertainty |
|---|---|---|---|---|
| 2-1-1A: Direct measurement: Input-output | IEC 60034-2-1 (IEC 60034-2-1, 2014) IEC 60034-2-3 (IEC 60034-2-3:2020, n.d.), (IEC TS 60034-30-2, 2016) | Six load points | Low | High uncertainty: primary contributing factor is the torque measurement |
| 2-1-1B: Summation of losses: Residual losses: Indirect (Loss segregation (Kärkkäinen, 2021)) | | | High | Low uncertainty: involves diverse test requirements. |
| 2-3-D: Calorimetric (IEC 60034-2-3:2020, n.d.), (Kärkkäinen, 2021), (Pagitsch et al., 2016) | | | Medium | Low uncertainty, high cost in building calorimetry system. |
| Alternative back-to-back (Zhang & Neshati, 2018) | Modified from (IEC 60034-2-1, 2014) | | Medium | Low uncertainty: requires device under test (DUT) operation as motor and generator, not feasible on all test benches. |
| Direct, indirect with variable speed drive (VSD) | IEC 61800-9-2 (IEC 61800-9-2, 2017) | Eight load points | Direct low, indirect high | Direct high, indirect low uncertainty |
| Iso-efficiency map (Weidinger, Foyer, Kock, Gnauert, & Kumme, 2018), (Weidinger et al., 2021a) | New load profile | Variable torque and speed load points | Medium | High uncertainty, but with ability to encompass a wider range of operating points |



## 2 Standardised efficiency determination of rotating electrical machines

When measuring the efficiency of electrical machines, there are two methods: direct and indirect. The direct method involves input/output measurements of both electrical and mechanical power, while the indirect method is based on measuring electrical power and power losses by summing them up (Kärkkäinen, 2021), (Pagitsch et al., 2016). The latter method is used to separate and identify the individual losses through various tests. More information on the principles of direct and indirect efficiency measurement methods can be found in (Zweiffel et al., 2021) in chapter 7.

### 2.1 Load steps and boundary conditions

Standardised methods for the efficiency determination of rotating electrical machines focus on a step-by-step measurement procedure of electrical $P_{elec}$ and mechanical power $P_{mech}$ at different load points. The variables to be measured are defined in IEC 60034-2-1 (IEC 60034-2-1, 2014) for motors with mains supply and in IEC 60034-2-3 [9] for motors driven by converters. Likewise, the load points are specified in IEC 60034-2-1 and IEC 60034-2-3 as percentages of the rated torque load of the

device under test (DUT). To characterise the efficiency of the DUT, at least six load points based on the rated load are required. A similar test method for power converters, described in IEC 61800-9-2 (IEC 61800-9-2, 2017), requires eight load points (Figure 1) related to the nominal load. This method, however, yields insufficient information about the MU, reproducibility, repeatability, and traceability of the efficiency determination. In order to execute the test methods mentioned, the drive unit must be able to generate at least 25 % more power than the full load of the DUT.

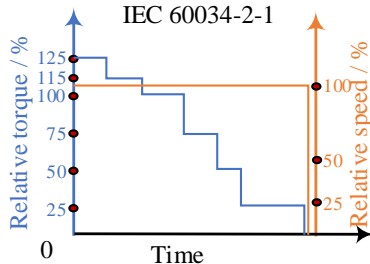

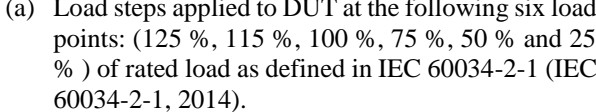

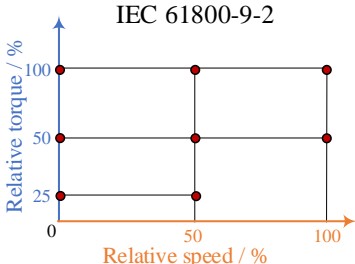

(a)   Load steps applied to DUT at the following six load points: (125 %, 115 %, 100 %, 75 %, 50 % and 25 %) of rated load as defined in IEC 60034-2-1 (IEC 60034-2-1, 2014).

(b)   Eight measurement points based on IEC 61800-9-2 (IEC 61800-9-2, 2017).

**Figure 1: Load profile used for efficiency determination with standardised (IEC) efficiency determination methods for rotating electrical machines.**

Furthermore, to reduce the influence of temperature on the DUT, the DUT is run at nominal load until thermal equilibrium is achieved (change rate 1 K or less per half hour) before the tests are carried out. The load is applied as quickly as possible to minimise temperature changes in the machine during testing.

The efficiency can be determined in motor as well as in generator mode as described in IEC 60034-2-1 (IEC 60034-2-1, 2014). In motor mode, the efficiency is determined by measuring the electrical input power ($P_{elec}$) and the mechanical output power



($P_\text{mech}$) directly. In the following, subscript $M$ is used for motor mode. In generator mode, on the other hand, the mechanical input power ($P_\text{mech}$) and the electrical output power ($P_\text{elec}$) are measured directly. For the generator mode, subscript $G$ is used:

$$\text{Motor mode} \qquad \eta_{\text{d\_M}} = \frac{P_\text{out}}{P_\text{in}} = \frac{P_2}{P_1} = \frac{P_\text{mech.M}}{P_\text{elec.M}} = \frac{2\pi \cdot n \cdot T}{\sqrt{3} \cdot U \cdot I \cdot \cos(\varphi)}, \qquad (1)$$

$$\text{Generator mode} \qquad \eta_{\text{d\_G}} = \frac{P_\text{out}}{P_\text{in}} = \frac{P_2}{P_1} = \frac{P_\text{elec.G}}{P_\text{mech.G}} = \frac{\sqrt{3} \cdot U \cdot I \cdot \cos(\varphi)}{2\pi \cdot n \cdot T}, \qquad (2)$$

where $n$ is the operating speed in rotations per minute (rpm), $T$ is the torque in N·m, and the electrical quantities are represented (as voltage $U$, current $I$, and power factor $\lambda = \cos \varphi$).

## 2.2 Measurement uncertainty calculation

In the following, methods complying with the Guide to the Expression of Uncertainty in Measurement (GUM) and using calibration certificates (CCs) are presented for the evaluation of MU when determining the efficiency of rotating electrical machines. In Figure 2, the GUM measurement model is presented in the form of a block diagram. The quantity of interest to be calculated is the electrical machine's efficiency, including its MU. This is calculated using the GUM based on the model equation given in IEC 60034-2-1 and applying various input quantities from tests and CCs of the instruments used. The CCs of the instruments and sensors used to measure the electrical, mechanical, and thermal quantities contain valuable information about the quality and measurement accuracy of the devices. When calibrating measurement instruments, they are traced to national standards, their performance and deviation from the national standard is evaluated, and the associated MU is determined. The data from the CCs are further interpolated to obtain the correct MU value for the relevant operating points, which is then fed into the model equation together with the corresponding probability distribution function (PDF). If no trend is discernible during the interpolation, larger MUs should be applied (e.g., if a certain value is measured, the neighbouring measured values should be considered and the larger MU values used). If a normal distribution with expanded MU ($k = 2$) is specified, a normal distribution is also used in the model equation; otherwise a rectangular distribution is used if no further information about the distribution is given.

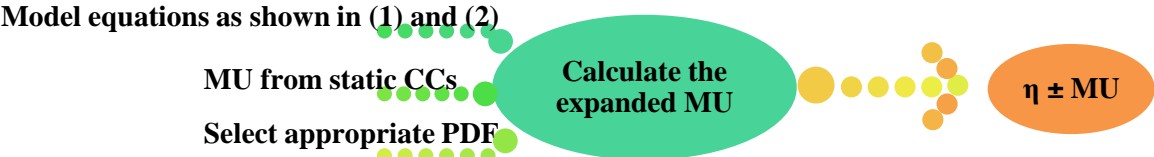

**Figure 2: GUM model for the expanded MU calculation of an efficiency determination.**

**Mathematical model of measurement uncertainty for direct efficiency determination based on IEC 60034-2-1**

Here, the uncertainty contributions for the direct efficiency determination method in generator mode are given. The calculation of the MUs is carried out with the information from the individually measured variables (electrical and mechanical quantities). The air pressure and relative air humidity are not taken into account. In order to ensure a coverage interval of 95 %, a coverage factor $k = 2$ is selected for the calculation of the expanded MU and a normal distribution is selected for the output variable





(direct efficiency). The combined (expanded) MU for the direct efficiency determination method of rotating electrical machines in generator mode with coverage factor $k = 2$ can be represented by (3).

$$MU\eta_{d_G} = k \cdot u_c(\eta_{d_G}) = k \cdot \sqrt{\left[\frac{\partial \eta_{d_G}}{\partial P_{elec.G}} \cdot u(P_{elec.G})\right]^2 + \left[\frac{\partial \eta_{d_G}}{\partial P_{mech.G}} u(P_{mech.G})\right]^2}, \quad (3)$$

where $MU\eta_{d_G}$ is the expanded MU for the direct efficiency determination method, $k = 2$ is a coverage factor for a normal distribution, and $u_c(\eta_{d_G})$ is the standard MU value.

The sensitivity coefficients of the efficiency for electrical output power $P_{elec.G}$ and mechanical input power $P_{mech.G}$ with partial derivatives are:

$$c_{P_{elec.G}} = \frac{\partial \eta_{d.G}}{\partial P_{elec.G}} = -\frac{P_{mech.G}}{P_{elec.G}^2}; \quad c_{P_{mech.G}} = \frac{\partial \eta_{d.G}}{\partial P_{mech.G}} = \frac{1}{P_{elec.G}}. \quad (4)$$

The uncertainty of $P_{elec.M}$ will have a small impact on the efficiency because the partial derivative $\frac{\partial \eta_{d.G}}{P_{elec.G}}$ is relatively small.

## 3 Iso-efficiency map


As mentioned above, standardised efficiency determination procedures are designed to determine efficiency using only a few (six to eight) load points. The so-called iso-efficiency contour or map method, on the other hand, is used to determine the efficiency over the entire operating range of rotating electrical machines. It was introduced by (Deprez et al., 2010), (Stockman et al., 2010), and (Vanhooydonck et al., 2010). In an iso-efficiency map, the efficiency is a function of rotational speed $n$ and

torque $T$, with rotational speed plotted on the abscissa and torque on the ordinate (Song et al., 2023). With this new method, the efficiency is determined directly by measuring all input and output variables. This approach was chosen because power loss segregation and determination in nacelle test benches (NTBs) are not possible due to the dimensions and the complexity of NTBs.

### 3.1 New load profile and measurement concept

The newly developed load profile for efficiency determination is based on standardised efficiency determination methods ((IEC 60034-2-1, 2014), (IEC 61800-9-2, 2017), (Weidinger et al., 2021a)) which use a combination of rotational speed $n$ and applied torque $T$. Similar to the approach outlined in the IEC standards (Figure 1), the novel profile also involves direct measurement of input and output parameters at six to eight load operating points. However, it distinguishes itself by applying six to eight torque levels at various rotational speeds (Figure 3) as opposed to a constant rotational speed (Figure 1 (a)).The

number of both rotational speed and torque levels depends on the total operating range of the DUT. The overall measuring range is specified accordingly by the DUT. As in the well-known standards, torque is always applied in descending order, i.e., from the largest to the smallest torque (as stipulated in IEC 60034-2-1). The rotational speed can be applied either in ascending or descending order; for these measurements, it was increased over time.




The new load profile is accurate, fast, and cost-effective with its full-range testing of wind turbines, including converter
performance. When there are no particular operating points for which the efficiency needs to be determined (such as typical
maximum power), the operating points should be distributed evenly. Attention must be paid to omitting operating points close
to the eigenfrequencies of the system, as this would lead to undesired dynamics or instable control effects. Due to the large
dimensions and the complexity of wind turbine drive trains, it is not possible within the limited measurement time frame to
bring the drive train into the thermal equilibrium state for the purpose of performing measurements. In order to analyse the
temperature influence on the efficiency, experimental measurements were carried out in an SSTB in the cold state and after
reaching thermal equilibrium.

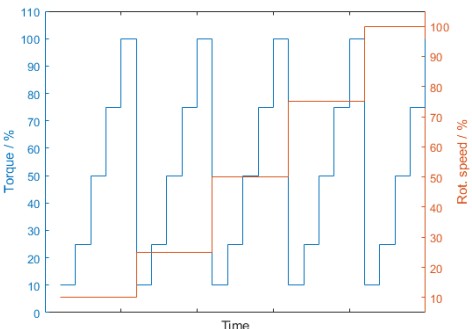 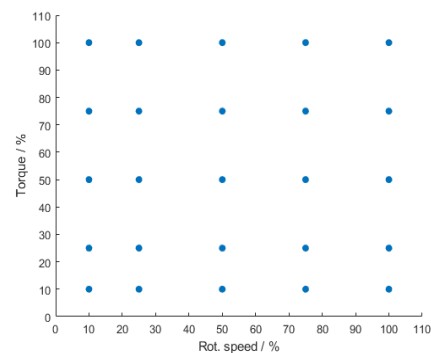

(a)   Iso-efficiency map load profile.                    (b)   Load point overview per operating range.

**Figure 3: Newly developed iso-efficiency map load profile** (Weidinger et al., 2018) **(a) and overview (b) of all measurement points adapted to the boundary conditions of the DUT used for efficiency determination and creating the iso-efficiency map.**

### 3.2 Torque zero signal determination

To measure the mechanical load correctly, the torque offset signal in the present mounting position with no load applied $T_c$
has to be determined and subsequently used to tare the measurement signal $T$. This is necessary because torque sensors, which
are based on the measuring principle of strain gauges, always show an offset. This zero signal can change slightly due to the
installation in the drive train and is position-dependent when installed horizontally. In order to obtain a suitable, position-
independent offset signal, the zero signal $T_i$ is recorded in at least four static positions ($m = 4$) evenly distributed over one full
revolution of the drive train, and the mean value of the four values is calculated. The same procedure is used in test benches
for wind turbines. A more detailed analysis of occurring influences and further explanations concerning the need to determine
the zero signal can be found in (Song et al., 2023).

$$T_{corr} = T - T_c,$$ (5)

with
$$T_c = \frac{1}{m} \sum_{i=1}^{m} T_i,$$ (6)

where $T_{corr}$ is the displayed torque value corrected for the offset value in the present setup, and $m$ is the number of static
positions evenly distributed over a full revolution of the drive train.



Another approach for signal nulling at minimum rotational speed in the positive and negative directions is described in (Zhang, Pieper, & Heller, 2023).

### 3.3 Signal evaluation

In the process of measuring electrical and mechanical quantities through signal analysis, it is imperative to attain a stationary state marked by minimised periodic effects. This condition must be met prior to conducting signal averaging for each operating
point on the test bench. For a traceable efficiency determination method, an influence study of torque and rotational speed ripples, as well as ripples in current and voltage measurement, should therefore be taken into account by calculating the operating point instability.

The establishment of traceability in the context of measuring electrical and mechanical quantities on test benches is crucial, especially when it comes to the determination of qualitative efficiency using designated MU values. The traceability of
measurements relies on loading conditions (as in Figure 1 and Figure 3) and signal processing methods. Notably, these methods differ for dynamic conditions (rotational conditions of torque and speed sensors) in a test bench. This interplay underscores a crucial aspect that demands a detailed discussion. The fundamental practice of traceable measurement of electrical and mechanical quantities on test benches is of utmost importance for maintaining the accuracy and reliability of efficiency measurements in this scientific research. This practice is proposed for detailed incorporation into this paper.

When measuring electrical and mechanical quantities, a stationary state (minimised periodic effects) must be reached before signal averaging per operating point is done. A previous work (Song et al., 2022) thoroughly details the stability of torque measurement points under rotary conditions on an SSTB for electrical machines. That paper highlights the torque measurement deviation caused by significant torque ripples, which serve as an additional source of uncertainty in the SSTB. In our research work, we have also adopted the stable load step methodology, which involves signal averaging per operating point and utilizes
a measurement instrument configuration similar to that outlined in the previous paper (Song et al., 2022) for rotating machines on the SSTB. Here, signal averaging in the SSTB is done over a constant time interval (minimum 8 s, here 10 s) and multiple revolutions and fragments of a revolution. This serves to average the signals over many more revolutions within the defined time intervals compared to the low-speed shaft of the DUT. Similarly, signal averaging in an NTB is also considered over an integer number of drive train revolutions when measuring torque on the LSS as described in (Song et al., 2023).

### 3.4 MU with operating point instability

For the torque step instability, the relative expanded MU is estimated for each load step based on the standard deviation $\sigma_T$ of 500 ms averaged values over a full 10 s averaging time interval:

$$MU_{T,inst} = \frac{2\sigma_T}{\eta t},$$ (7)

where $\eta t$ is 500 ms averaged values over a full 10 s averaging time interval.





The dynamic instability and oscillations in torque exert an immediate influence on the rotational speed. Given the inherent correlation between torque variations and rotational speed changes, these oscillations do not make a direct contribution to the computation of the relative expanded MU in mechanical power measurements:

$$MU_{\text{mech,inst}} \neq \sqrt{MU_{\text{T,inst}}^2 + MU_{\text{n,inst}}^2} \,. \tag{8}$$

Instead, the MU of the operating point instability of the efficiency $MU_{\eta,\text{inst}}$ should be estimated by observing the oscillation of the efficiency itself using the same method as in (7).

As discussed earlier, the assessment of torque, rotational speed, and operating point instability collectively contributes to the overall relative expanded MU inherent in the determination of mechanical power as:

$$MU_{\text{mech}} = \sqrt{MU_{\text{T}}^2 + MU_{\text{n}}^2 + MU_{\text{mech,inst}}^2} \,. \tag{9}$$

The dynamic influence of torque instability and oscillation on rotational speed prompts variations in the generated electrical power. This causes similar instabilities in these measured parameters. Given the correlated nature of oscillations among torque $T$, rotational speed $n$, voltage $U$, and current $I$, their direct contribution to the calculation of relative expanded MU in efficiency measurement is limited.

## 4 Experimental setups

To validate the newly developed iso-efficiency map method with respect to efficiency results and their associated MUs, the method was applied on a four-pole rotating electrical machine tested on a 200 kW SSTB. It was then carried out in a 4 MW NTB, where it is to be used in the future to determine the efficiency of wind turbine drive trains.

### 4.1 200 kW small-scale test bench

The 200 kW SSTB is owned by the Physikalisch-Technische Bundesanstalt (PTB) in Braunschweig, Germany. PTB is Germany's national metrology institute.

### 4.1.1 Experimental setup with DUT and sensors

A small-scale 200 kW motor test bench with two different torque transducers and a test rotating electrical machine (Figure 4) was used to investigate the newly developed iso-efficiency map method and to validate it by comparing the efficiency with the results of the standardised direct efficiency determination method performed in the same setup. To this end, a four-pole rotating electrical machine, which is an ASM, was used as the DUT. It was operated at various load conditions with both varying torque and varying rotational speed. The specifications of the DUT and the variable speed drive (VSD) are summarised in Table 2 and 3, respectively.

The entire efficiency determination process for rotating electrical machines, including thermal stability measurements, is monitored and controlled by a real-time test control system. During the test, the stator winding temperature, the surface



temperature of the DUT, and the ambient temperature are continuously measured by temperature sensors (Pt100 sensors and type T thermocouples). Prior to starting the measurements, the DUT is run continuously until thermal stability is reached. This thermal stability is achieved when the rate of temperature change at the hottest point is less than 1 K per half hour. The surface temperature is measured by a temperature recorder.

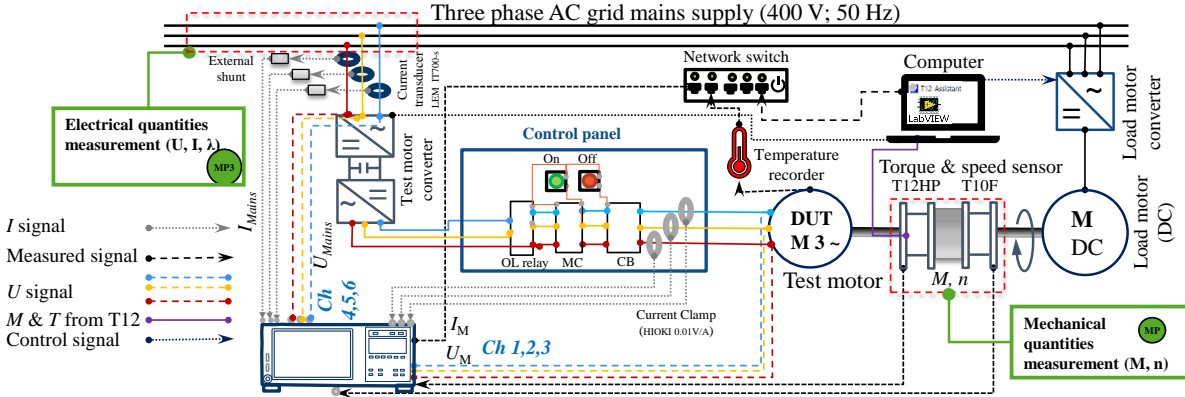

(a)  Schematic diagram of the SSTB.

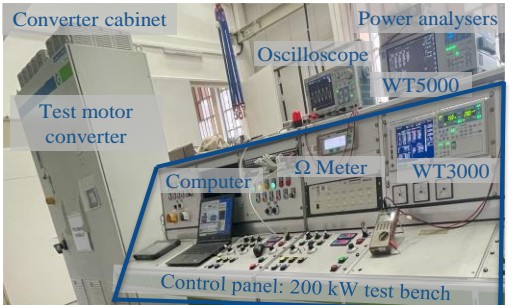

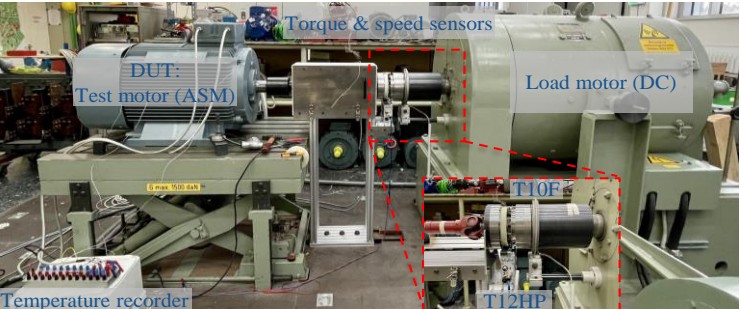

(b)  Photographs of the SSTB, with the left image displaying the control panel (including the converter and measurement devices), and the right image depicting the DUT and load machine equipped with torque and rotational speed sensors.

**Figure 4: Experimental setup for the efficiency measurements of rotating electrical machines with the layout of the measurement**
**sensors on the SSTB.**

**Table 2: Rated nameplate values of the ASM DUT used on the SSTB at a rated frequency of 50 Hz.**

| | Rated nameplate values | | | | | | |
|---|---|---|---|---|---|---|---|
| | Power / kW | Torque / kN·m | Speed / rpm | Voltage / V | Current / A | Power factor $\lambda$ | Remarks |
| ASM1 | 160 | 1.028 | 1485 | 400 | 294 | 0.82 | SSTB |

**Table 3: Rated nameplate values of the converter (VSD) used in the SSTB.**

| Power / kW | O/p $I_{nom}$ / A | O/p $U_{nom}$ / V | O/p $f$ / Hz | Mains voltage / V | $f_{sw}$ / kHz |
|---|---|---|---|---|---|
| 200 | 375 | $(0 - 1.2) \, x$ mains voltage | $0 - 400$ | 380-460 + 10 % / -15 % | 3 |

All electrical quantities (such as voltage $U$, current $I$, and power factor $\lambda = \cos \varphi$) used to determine the electrical power $P_{elec}$,
and the mechanical quantities (such as torque $T$ and rotational speed $n$) used to determine the mechanical power $P_{mech}$, are



measured by the Yokogawa WT 5000 power analyser and read out on a computer. The instruments and sensors used are summarised in Table 4. The mechanical quantities of torque $T$ and rotational speed $n$ are measured using an HBK T12HP with pulse outputs for both measurement variables. The electrical current of all three phases is measured using LEM IT700-s current sensors via the WT 5000. Similarly, the three phase voltages are measured by the Yokogawa WT 5000 power analyser, which

directly taps them in parallel as shown in Figure 4 (a). This method allows us to obtain accurate voltage readings during the experiments, given that there are no further sensors available in the voltage measurement chain.

**Table 4: Measuring instruments and sensors used on the SSTB to gather mechanical, electrical and thermal quantities.**

| Devices and sensors | Quantities | | | | | | |
|---|---|---|---|---|---|---|---|
| | Mechanical | | Electrical | | | Thermal | |
| | Torque $T$ | Speed $n$ | Voltage $U$ | Current $I$ | $\lambda$ | Stator winding temperature | Surface temperature |
| Data acquisition system | Power analyser: Yokogawa WT 5000 | | | | | LR12000E | |
| Sensors in SSTB | HBK T12HP | | - | LEM IT700-s | - | Type T thermocouples | |

### 4.1.2 Zero signal determination through static torque offset measurement in the SSTB

Torque transducers have a signal offset caused by various factors. To account for this offset, measurements are taken at eight

evenly distributed positions during one full revolution (torque $T_i$). The resulting static offset $T_c$ is subtracted from the measured torque $T$ to recalculate the torque signal $T_{corr}$ in post-processing and thus ensure accurate measurement results.

**Table 5: Measured torque value for T12HP at eight different positions of SSTB.**

| Position | 1 | 2 | 3 | 4 | 5 | 6 | 7 | 8 |
|---|---|---|---|---|---|---|---|---|
| $T_i$ | 0.096 | 0.091 | 0.069 | -0.005 | 0.114 | 0.141 | 0.200 | 0.168 |
| $T_c$ | | | | 0.109 | | | | |
| $T_{corr}$ | | | | $T - 0.109$ | | | | |

### 4.2 4 MW nacelle test bench

At the Chair for Wind Power Drives (CWD) of RWTH Aachen University, onshore wind turbines can be tested on a 4 MW

NTB. The NTB (Figure 5) employs a low-speed permanent magnet synchronous machine as a direct drive prime mover to handle high dynamic torque loads and rotational speeds up to 30 rpm. The maximum torque that can be applied on the NTB is 3.4 MN·m. The non-torque loading (NTL) unit, located between the prime mover and the DUT, is a servo hydraulic system that provides forces in three and bending moments in two directions. The NTB has its own torque transducer installed on the LSS between the prime mover and the NTL unit, but it was not used for efficiency determination in this measurement campaign

due to losses caused by frictions in the NTL unit.





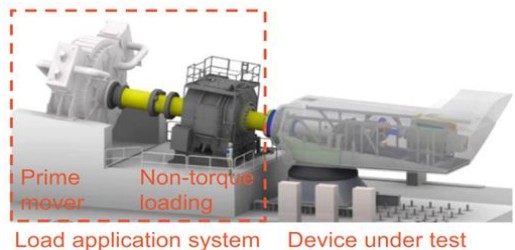

**Figure 5: NTB at the CWD of RWTH Aachen University, including FVA nacelle. Modified based on** (Kock, Jacobs, & Bosse, 2019)**.**

### 4.2.1 Experimental setup with DUT and sensors

The DUT is a 2.75 MW research nacelle provided by FVA (Forschungsvereinigung Antriebstechnik e. V.) and operated by
CWD. It is a wind turbine nacelle, specifically a NEG MICON NM80, and its performance data are listed in Table 6. The FVA research nacelle's drive train includes a main bearing that supports the rotor, a gearbox that alters the speed, a single-fed induction generator with six poles that transforms mechanical energy into electrical energy, and a full-size converter concept (Figure 6). The gearbox comprises a planetary gear stage and two helical gear stages, enabling generator speeds up to 1100 rpm. A comprehensive visual representation of the NTB is provided through photography in Figure 6, with the left image displaying
the $T$ and $n$ sensors, the middle image showcasing the DUT (including the converter and measurement devices as well as the load machines equipped with gears), and the right image illustrating the voltage and current transducers on the grid side.

**Table 6: Performance data of the 2.75 MW research nacelle** (Song et al., 2023)**.**

| Quantity | Nominal power | Rotor diameter | Res. swept area | Rated rotor speed | Rated torque | Gear ratio |
|---|---|---|---|---|---|---|
| Value | 2750 kW | 80 m | 5027 m$^2$ | 17.5 rpm | 1550 kN·m | 62.775 |

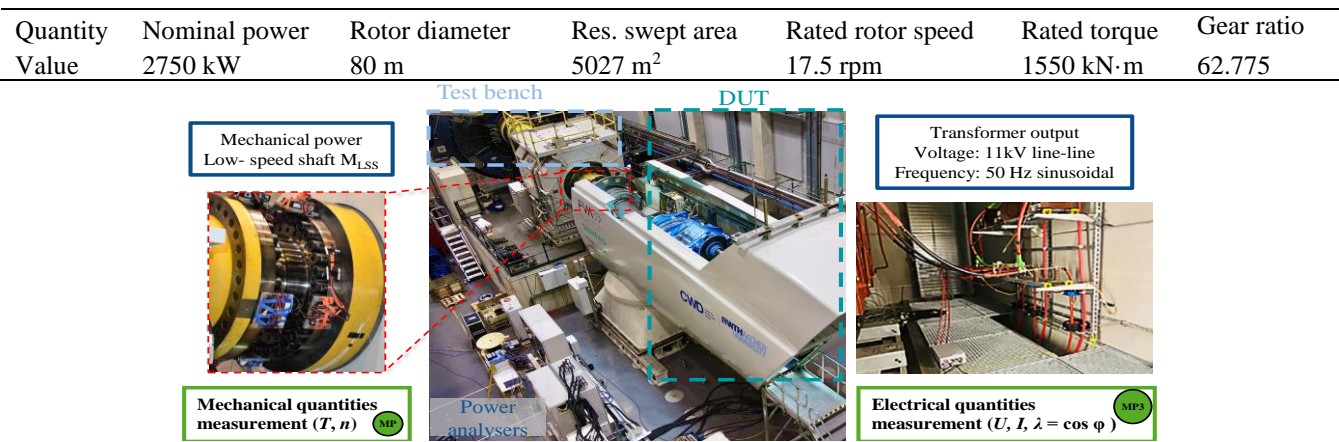

**Figure 6: Photographs of the experimental setup for efficiency measurements of a 2.75 MW wind turbine drive train on the 4 MW**
**NTB of CWD at RWTH Aachen University. The mechanical input power is measured on the LSS (left) and the electrical output power at the grid interface (right) (based on** (Weidinger et al., 2021a), (Song et al., 2023))**.**

### 4.2.2 NTB zero signal determination through static torque offset measurement in the NTB

The static zero offsets were computed using the averaging process outlined in equation (6). Torque averaging was conducted by marking the shaft at 12 positions separated by 30° intervals and incrementally rotating the shaft in the operational direction





to each position and holding the shaft stationary there for 60 seconds. A comprehensive explanation of this procedure is provided in the referenced paper (Song et al., 2023).

## 5 Efficiency determination results with MU and comparison

This section presents the results of direct efficiency determination using standardised IEC methods and the newly developed iso-efficiency map. Additionally, a comparative analysis of the results is conducted, accompanied by supplementary insights
into the MU, including the influences of operating point instability.

### 5.1 Direct efficiency measurement results obtained with standardised methods in motor and generator mode

This section offers a brief overview of the results obtained through direct efficiency determination and considering the MU in accordance with IEC 60034-2-1. In (Yogal, Lehrmann, & Zhang, 2022), a thorough analysis of efficiency determination results is presented that incorporates the MU. This comprehensive study includes a comparative assessment of direct, indirect, and
alternative back-to-back efficiency determination methods with an evaluation of the MU.

### 5.1.1 Direct efficiency measurements in generator and motor mode with mains supply (without converter)

Following the guidelines provided in IEC 60034-2-1, measurements of six load points at thermal equilibrium were conducted in both generator and motor mode. The measured temperature curves are shown in Figure 7. However, it is evident that the six load points in generator mode do not align with those in motor mode, deviating from the specifications of IEC 60034-2-1. This
deliberate deviation was employed to ensure equal electrical power in both generator and motor modes since the alternative back-to-back method was used, which requires equal electrical power in both operation modes.

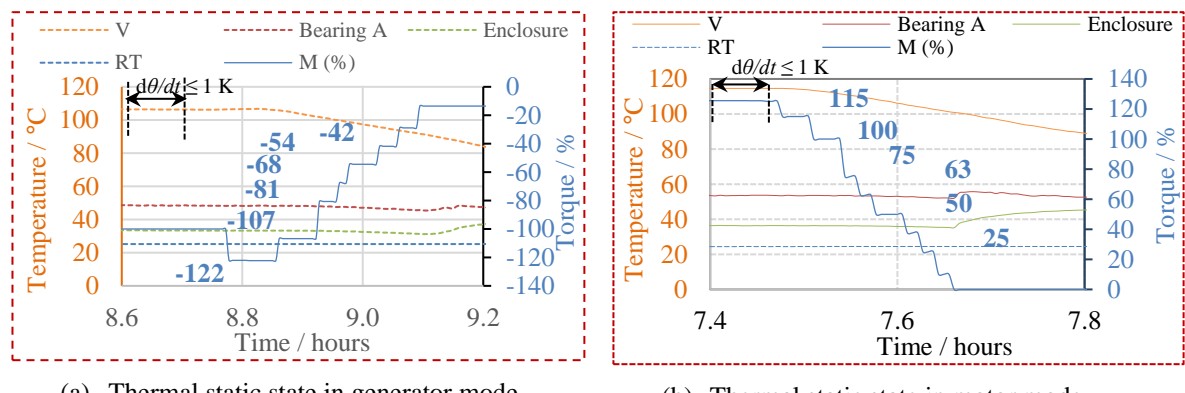

(a)    Thermal static state in generator mode.          (b)    Thermal static state in motor mode.

**Figure 7: Measured temperature curves including steady state sequences obtained with standardised methods for efficiency measurements in generator (a) and in motor (b) mode at rated speed.**





Figure 8 presents the outcomes of the direct efficiency measurement based on equations (1) and (2) in both generator and motor mode and including the relative expanded MU of the efficiency determined by equation (3) at six distinct load points. In this section, only the direct method is included for analysis. The results of the indirect and the alternative back-to-back efficiency determination methods are presented in (Yogal et al., 2022). This paper focuses only on the direct efficiency determination in generator mode according to existing standards and to the newly developed iso-efficiency map method. The

calculated values for the direct efficiency determination method with MU for rotating electrical machines are presented with a normal distribution, i.e., a coverage factor of $k = 2$.

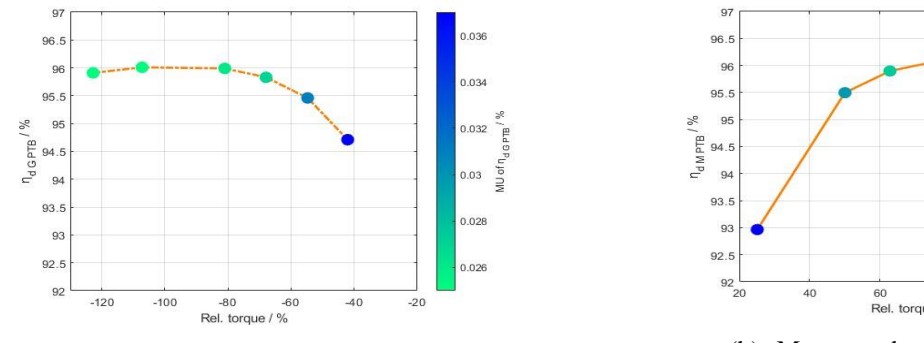

(a) Generator mode.

(b) Motor mode.

**Figure 8: Direct efficiency measurements in generator (a) and motor (b) mode, including relative expanded MU at rated speed** (Yogal et al., 2022) **based on IEC 60034-2-1** (IEC 60034-2-1, 2014)**.**

**5.1.2 Direct efficiency measurements in generator mode with feed to grid (with converter)**

Examples of DUT temperature curves, including steady state sequences (thermal stability), for the IEC 60034-2-1 measurements in generator mode with feed to grid via a converter are presented in Figure 9. During all measurements, the rate of temperature change was less than 1 K per half hour at the hottest point as stipulated in IEC 60034-2-1. In the WT 5000 power analyser, mechanical parameters such as rotational speed and torque and electrical parameters such as current, voltage, and the power factor were recorded synchronously. Using the WT 5000 motor evaluation function, mechanical and electrical

power as well as the DUT efficiency were calculated. For all different load profiles (standardised methods), it was ensured that the thermal steady state of the DUT was reached.



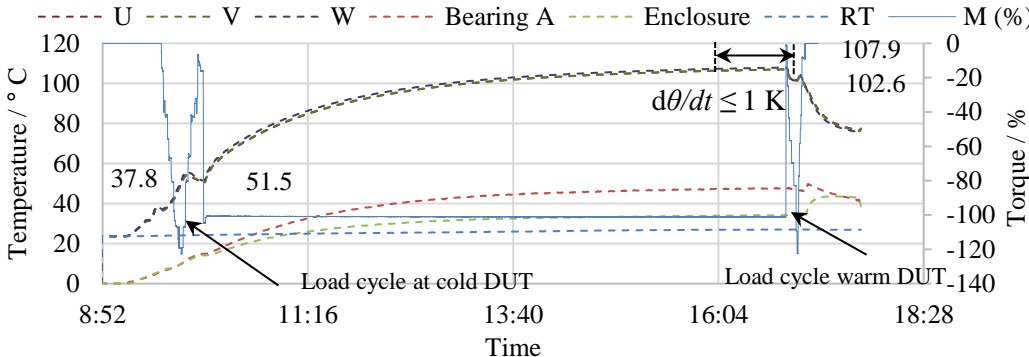

**Figure 9: Measured temperature curves including load cycle for measuring DUT efficiency in generator mode with feed to grid (with converter).**

In Figure 10, the direct efficiency measurement results are depicted, including relative extended MU values in accordance with

IEC standards. These measurements were carried out in the SSTB in both cold (a, b) and warm (c, d) conditions in generator

mode while connected to the grid via a converter.

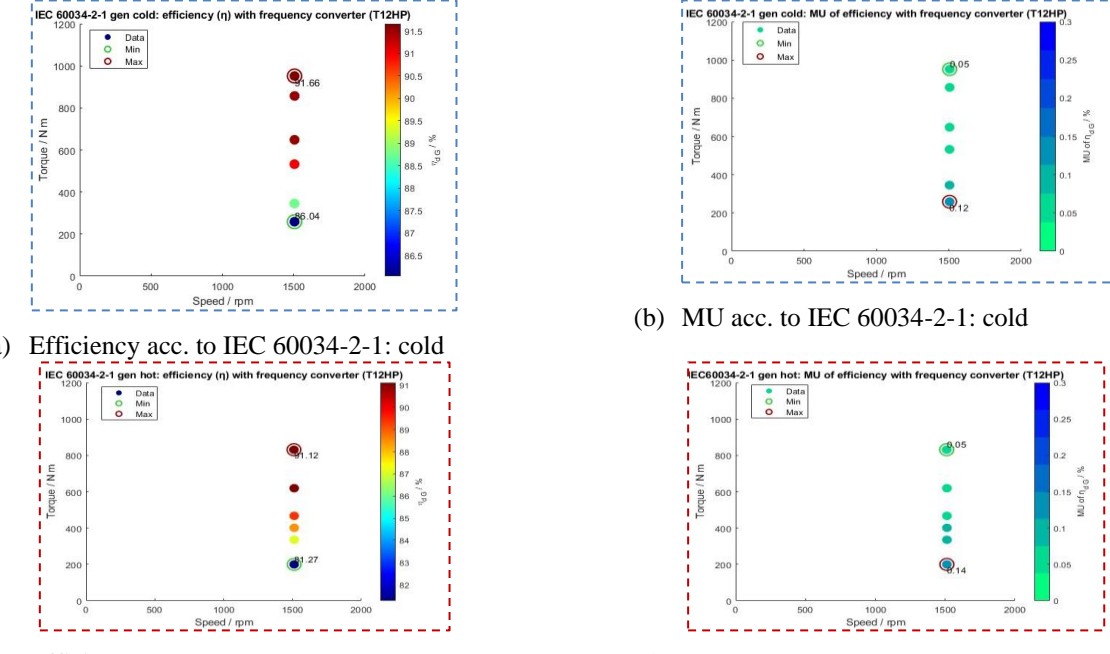

(a) Efficiency acc. to IEC 60034-2-1: cold

(b) MU acc. to IEC 60034-2-1: cold

(c) Efficiency acc. to IEC 60034-2-1: warm

(d) MU acc. to IEC 60034-2-1: warm

**Figure 10: Direct efficiency measurement results including relative expanded MU acc. to IEC standards in the SSTB in cold (a, b) and warm (c, d) generator mode – feed to grid via a converter.**

The efficiency and the relative expanded MU determined in the warm state of the DUT (Figure 10 (c) and (d)) are lower

compared to those measured on a cold DUT (Figure 10 (a) and (b)). This trend can be attributed to increased resistive losses

and elevated operating temperatures in the warm DUT scenario, which adversely affect the overall efficiency. During warm

generator mode, the efficiency decreases as the load reduction increases, pointing to a typical trend in power systems. The




observed efficiency values $\eta_{d\_G\_IEC\_wFU}$ range from approximately 81.27 % to 91.12 %, with an associated expanded MU, $MU_{\eta d\_G\_IEC\_wFU}$, spanning from 0.047 % to 0.141 %. This uncertainty signifies the potential variability in the measurements

due to factors such as instrumentation precision and experimental conditions. Such detailed analysis underscores the intricacies inherent in efficiency measurements and highlights the importance of considering uncertainty when interpreting efficiency results.

**5.2 Measurement results with the iso-efficiency map method**

Figure 11 (a) presents a comprehensive depiction of the load profile used to create the iso-efficiency map for the DUT on the

SSTB. This figure helps to demonstrate how the efficiency changes with changes in electrical and mechanical power. It also shows the losses when the number of operating points decreases. The measured values obtained in this study provide valuable insights into the efficiency characteristics of the DUT and the system as a whole. By enabling accurate efficiency determination through interpolation at any operating point, they serve to improve the evaluation process. Additionally, Figure 11 (b) visually depicts the correlation between torque and rotational speed at 43 operating points, offering guidance for informed decisions

regarding the DUT's efficiency in various operating scenarios.

The iso-efficiency map offers the advantage of allowing easy observation of the impact of speed on torque fluctuations during speed transitions. In the given operating point map, measurements were taken for over 40 s per operating point, with stable torque and rotational speed signals being averaged over 10 s after a waiting time of 20 s to determine the efficiency.

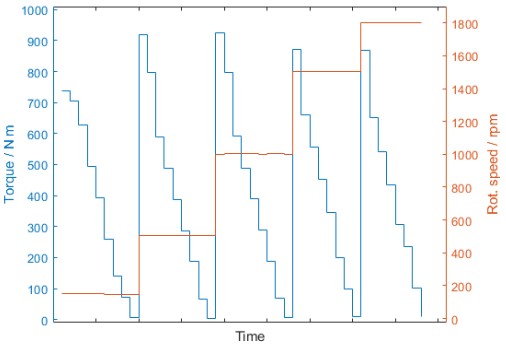
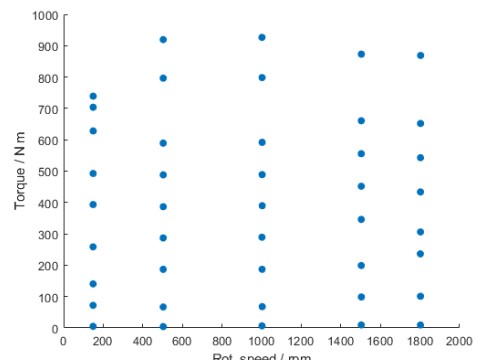

(a) Iso-efficiency map load profile.                    (b) Load point overview per operating range.

**Figure 11: Iso-efficiency map load profile (a) and overview (b) of all 45 measurement points adapted to the boundary conditions of**

**the DUT used for efficiency determination and for creating the iso-efficiency map.**

To analyse the effect of machine temperature on efficiency determination, the iso-efficiency map measurements were performed twice in generator mode: once in the cold and once in the warm state (with thermal stability) of the DUT. As can be deduced from the machine temperature curves in Figure 12 (a), the difference between the cold and warm states of the machine is approx. 75 K. Being in the cold state also means that the machine is not in thermal equilibrium. This in turn means

that the winding resistance is lower than when the entire machine, including its windings, is warmed up.



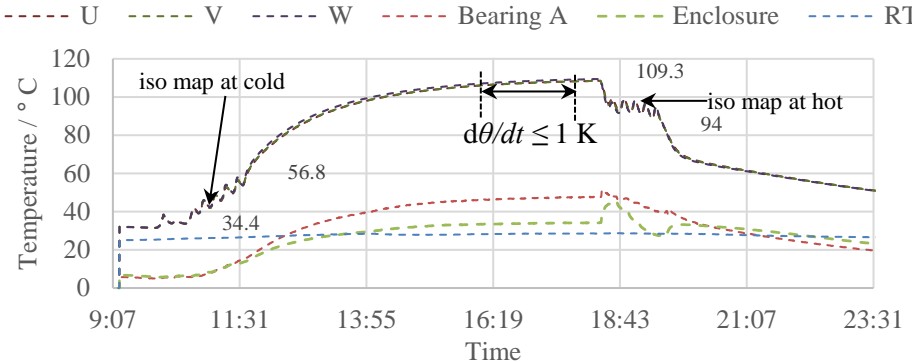

**Figure 12: Measured temperature curves of the generator stator windings and the measured mechanical parameters of torque *T* and rotational speed *n* for the iso-efficiency map measurements in thermal steady state on the SSTB.**

### 5.2.1 Direct iso-efficiency map measurements in generator mode with feed to grid (with converter)

In this study, the efficiency of a single generator was measured at 43 iso-efficiency map operating points in two different
operating conditions: cold and warm generator. These two measurements enable the direct efficiency calculation of the generator under varying temperatures of the system (considering only the DUT temperature as depicted in Figure 12 (a)). The results again show that the generator's efficiency when cold (ranging from about 18.6 % to 92.1 %) is actually higher than that of the warm generator, where the range of efficiencies is lower at about 15.7 % to 91.9 %. This again indicates that the operating temperature of the generator has a significant impact on its overall efficiency. The decrease in efficiency observed
when the generator is warm could be due to other factors, such as losses associated with higher temperatures, in addition to the increased resistance.

To obtain accurate efficiency measurements on a test bench, it is therefore crucial to conduct these measurements after the generator has reached thermal steady state conditions. This means that sufficient time must be allowed for the generator temperature to stabilise to ensure consistent operating conditions in the DUT and also in the measurement sensors and devices.
By measuring efficiency under thermal steady state conditions, the effects of temperature-related factors, such as increased resistance and losses, on the generator's performance can be effectively assessed.





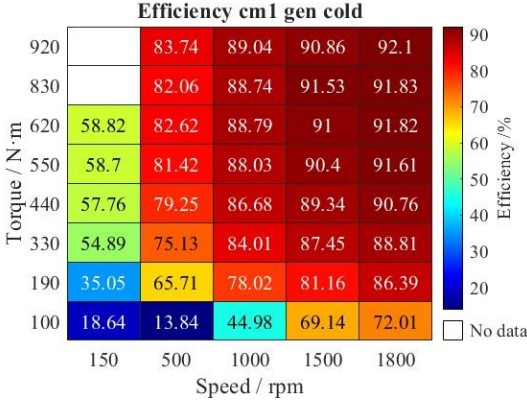
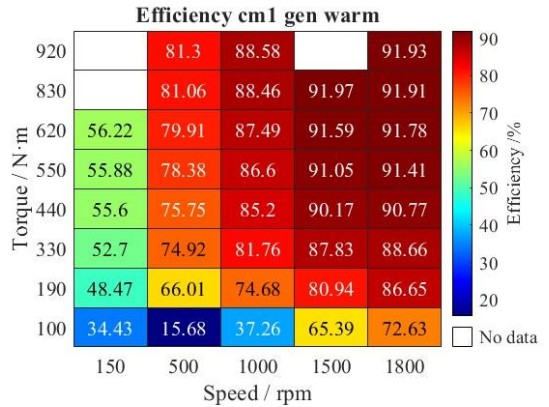

(a) Iso-efficiency map for cold generator.  (b) Iso-efficiency map for warm generator.

**Figure 13: Iso-efficiency map of direct measurements of DUT in generator mode on the SSTB: (a) in the cold generator state and (b) after the stator windings reached thermal steady state.**

In addition, the direct efficiency determined for a thermally stable (warm) generator is illustrated for the case without converter

(Figure 14 (a)) and the case with converter (Figure 14 (b)). As can be seen, the efficiency is highest at 91.968 % in the operating point at nominal torque (~800 N·m) and nominal rotational speed (1500 rpm). Moreover, variations in efficiency contours, ranging from cold to hot, can be observed due to temperature changes. However, it is challenging to see these variations in Figure 14. As was to be expected, the overall efficiency of the generator setup with frequency converter is lower (91.99 %) than without the frequency converter (96.08 %).

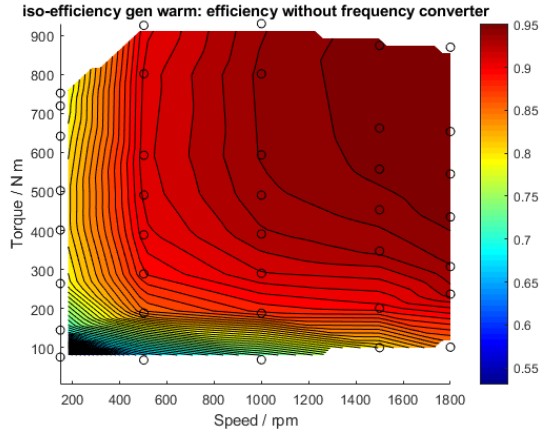
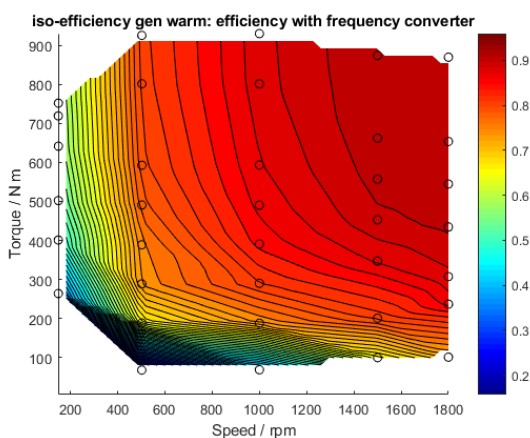

(a) Iso-efficiency map without frequency converter.  (b) Iso-efficiency map with frequency converter.

**Figure 14: Iso-efficiency map in warm mode without (a) and with (b) frequency converter.**

### 5.2.2 Iso-efficiency map results with MU based on static CCs

As discussed in the previous sections, the evaluation of the efficiency of the DUT equipped with a frequency converter involves measuring its mechanical input power and electrical output power on the grid side. To assess the performance of the complete system (generator and frequency converter) tested on the SSTB, the iso-efficiency map method was applied, and the relative





expanded MU of the entire DUT was also calculated (Figure 15). The results show an overall efficiency of about 91.86 % (cold generator) and 91.99% (warm generator) in the high torque and high rotational speed regime. For the very low torque and rotational speed ranges, on the other hand, the efficiency ranges from 13.89 % (cold generator) to 15.73 % (warm generator). In particular, it has been observed that in the lower torque and lower rotational speed ranges, a marginally higher overall MU of 0.27 % (cold generator) and 0.26 % (warm generator) occurs. However, at the DUT's rated torque and rated

rotational speed, the MU values of the entire system tested on the SSTB decrease to about 0.05 %, as shown in Figure 15 (b).

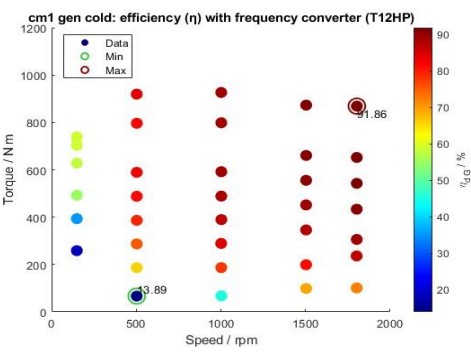

(a)   Efficiency with frequency converter in cold mode.

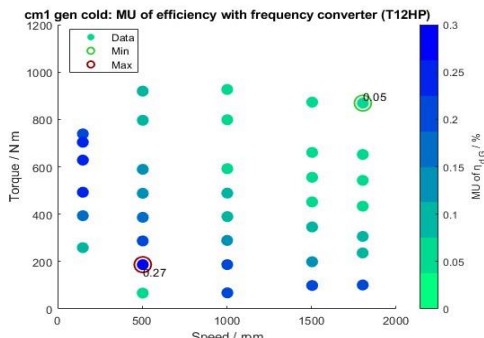

(b)   MU for efficiency with frequency converter in cold mode.

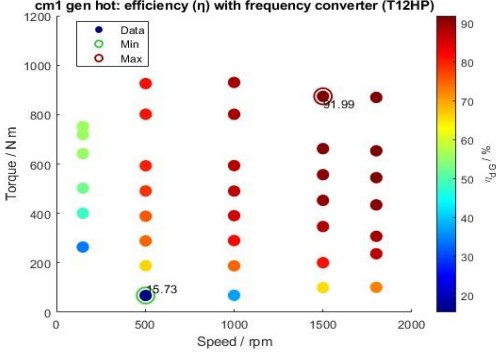

(c)   Efficiency with frequency converter in warm mode

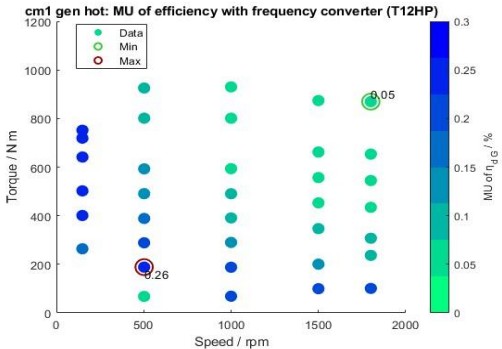

(d)   MU for efficiency with frequency converter in warm mode.

**Figure 15: Iso-efficiency map and MU overview considering only values from static CCs in cold (a, b) and warm (c, d) generator mode with frequency converter.**

### 5.2.3 Iso-efficiency map of MU including operating point instability

This section discusses the MU and also considers the dynamic operating point instability. The basic theory has already been

explained in section 3.4. The direct efficiency uncertainty model for measurements on the SSTB is expanded to include the uncertainty contribution from the operating point instability.

Up to this point in the paper, investigations have primarily focused on conventional efficiency measurements with MU, considering only the static calibration values of the sensors used and ignoring the dynamic rotation of the DUT. In contrast, this section addresses efficiency determination and thus also torque measurement under existing rotation conditions that result





in complexities such as ripples and increased signal noise. Due to the rotational motion, measurements are also subject to disturbances, which can in turn lead to instabilities in the torque measurement. A comprehensive analysis of these challenges has been extensively documented in (Song et al., 2023).

In the previous sections, static MU values ($MU_{\text{static}}$) are represented as $MU\eta_{\text{d}_\text{G}}$. In the following, efficiency determination with MU values is shown taking into account the instability of the electrical and, especially, the mechanical power measurement

under rotation. Accordingly, the overall MU now also includes the operating point instability ($MU_{\text{inst}}$) in addition to the static calibration values ($MU_{\text{static}}$). The relative expanded MU associated with determining the system efficiency consists of the static uncertainty contributions of torque, rotational speed, electrical power, and operating point instability. This is calculated as follows:

$$MU\eta_{\text{d}_\text{G (total)}} = \sqrt{MU_{\text{static}}^2 + MU_{\text{inst}}^2}. \qquad (10)$$

The relative expanded MU values which consider static calibration results and the operating point instability via the signals

for the efficiency determination of the generator with frequency converter in warm mode are presented in Figure 16. It shows the graphical representation of each component's contribution from equation (10), including operating point instability-related uncertainty and static MU. Of these contributions, the low values of the torque measurement (about 10 %) represent the largest source of uncertainty, contributing to an overall uncertainty of 0.79 %. Conversely, a lower overall MU of 0.05 % can be observed in the higher torque and higher rotational speed ranges.

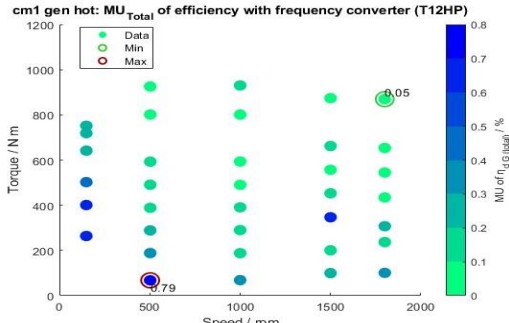

**Figure 16: Relative expanded efficiency MU (%) considering operating point instability and static MU values.**

Across the majority of measurement points in Figure 16, the MU values exhibit a consistent trend (higher MU at lower torque range). At 1500 rpm, an overall MU of 0.66 % particularly stands out, but not, as suspected, at the lowest torque level. Further investigation of the signals is advisable in order to clarify the primary causes. The higher MU of 0.66 % at 40 % torque can mainly be attributed to the instability of the original measurement signals. Given the direct relevance of the larger MU for

determining the system efficiency, a detailed analysis is recommended.

Figure 17 presents a characteristic instability in the original measurement signal of electrical power and efficiency at 1500 rpm and 40 % torque conditions. The electrical power (elec. power wFU, 1500 rpm) fed into the grid via the inverter was unstable



at the operating point (40 % torque values, interval start and end marked), as shown in Figure 17 (a). Due to unstable electrical signals, the efficiency of the system becomes unstable (88.79 % to 87.31 %) as shown in Figure 17 (b).

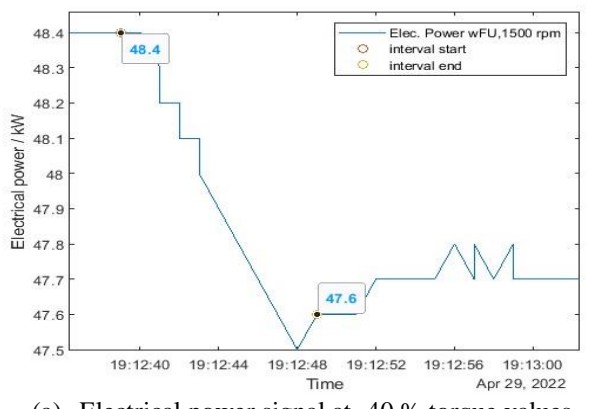
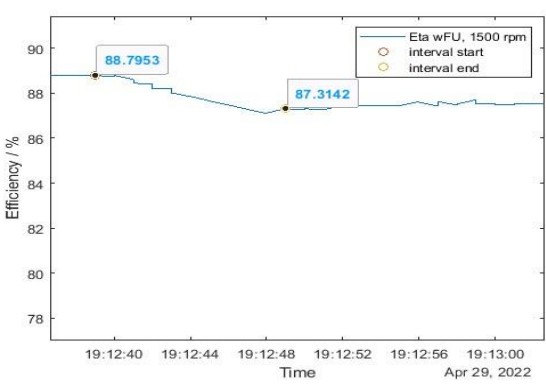

(a) Electrical power signal at -40 % torque values.

(b) Calculated efficiency at -40 % torque values.

**Figure 17: The original operating point instability for electrical power and efficiency values for 1500 rpm and -40 % torque values.**

Analysis of the DUT's iso-efficiency map on the SSTB shows optimum performance within its nominal operating range (as shown in Table 2). Notably, instability increases significantly beyond these operating points (exemplified by Figure 17 (a)), with operating points outside the nominal range exhibiting pronounced instability. The analysis of instability is thus a remarkable addition to the overall MU. Consequently, when performing efficiency measurements, due attention should be given to the instability associated with the operating point. This is particularly important in the case of wind turbine drive trains, since these have a larger nominal operating range in the field and are often operated outside of this operating range, depending on the prevailing conditions and the type of control over the wind turbine.

### 5.3 Measurement uncertainty contributions in the SSTB

This section deals with the uncertainty contribution of all measured parameters essential for determining efficiency. Ultimately, the combination of these results will yield the comprehensive total MU linked to the determination of the overall system efficiency.

### 5.3.1 Measurement uncertainty contribution from mechanical quantities

This section provides a comprehensive overview of the contributions to the MU associated with torque $T$ and rotational speed $n$ in the study for direct efficiency measurement. Table 7 is relevant to understand the accuracy and reliability of mechanical power measurement without offset error $P_{2corr}$. It presents detailed information on the sources of uncertainty related to rotational speed and torque measurements, allowing for a comprehensive assessment of the overall MU in the experimental setup. By quantifying the individual contributions from these mechanical quantities, valuable insights into the reliability and robustness of the measurements performed in this study can be gained. The results show that the torque measurement





contributes most (97.2 %) to the overall MU at a single operating point (rated torque and speed values). Compared to torque,
the rotational speed measurement makes a very small contribution (2.8 %) to the overall relative MU.

**Table 7: Relative expanded uncertainty contributions from torque and speed measurements of the DUT.**

| Quantity | Value | Standard uncertainty | PDF | Sensitivity coefficient | Uncertainty contribution | Index |
|---|---|---|---|---|---|---|
| $T$ | -857.766 N·m | $39.2 \cdot 10^{-3}$ N·m | Normal | 160 | 6.2 W | 97.2 % |
| $T_c$ | $-582.0 \cdot 10^{-3}$ N·m | | | | | |
| $T_{corr}$ | -857.184 N·m | $39.2 \cdot 10^{-3}$ N·m | | | | |
| $n$ | $1.5123 \cdot 10^3$ rpm | $11.7 \cdot 10^{-3}$ rpm | Normal | -90 | -1.0 W | 2.8 % |
| $P_{2corr}$ | $-135.748 \cdot 10^3$ W | 6.30 W | | | | |

### 5.3.2 Measurement uncertainty contribution from electrical quantities

When measuring electrical quantities (current, voltage, and power) that are fed into the grid, greater accuracy can be achieved
than when measuring mechanical quantities. This is due to the higher precision of current sensors and power analysers as
shown in Table 4. The measurement chain for electrical quantities primarily involves external current sensors and a power
analyser. Consequently, the MU values are relatively low in comparison to mechanical quantities, as shown in Table 8.

**Table 8: Relative expanded uncertainty contributions from electrical quantity measurement values of DUT.**

| Quantity | Value | Standard uncertainty | PDF | Sensitivity coefficient | Uncertainty contribution | Index |
|---|---|---|---|---|---|---|
| $I$ | 242.860 A | $28.6 \cdot 10^{-3}$ A | | | | |
| $I_1$ | 244.446 A | $50.0 \cdot 10^{-3}$ A | Normal | -180 | -8.9 W | 31.9 % |
| $I_2$ | 238.225 A | $48.4 \cdot 10^{-3}$ A | Normal | -180 | -8.7 W | 30.0 % |
| $I_3$ | 245.909 A | $50.3 \cdot 10^{-3}$ A | Normal | -180 | -9.0 W | 32.4 % |
| $U$ | 398.988 V | $11.5 \cdot 10^{-3}$ V | Normal | -330 | -3.8 W | 5.7 % |
| $\lambda$ | $-776.553 \cdot 10^{-3}$ | | | | | |
| $P_1$ | $-130.332 \cdot 10^3$ W | 15.8 W | | | | |

### 5.4 Comparison: standardised methods (IEC) vs. iso-efficiency map with static MU values

Directly comparing standardised methods with iso-efficiency maps poses challenges due to the inherent differences involved.
Standardised methods typically have a limited number of operating points at a thermally stable state of the DUT, resulting in
a winding temperature variation of approximately 4.9°K (Figure 9 (a)). Conversely, the iso-efficiency map method uses a
larger number of operating points at a nearly thermally stable state of the DUT over a longer time period, resulting in a larger
winding temperature variation of about 15.3°K (Figure 12 (a)). In this context, a comparison between standardised methods
and the iso-efficiency map method specifically at the rated speed values (100 %) is presented and depicted in Figure 18.
In the comparison between standardised methods (IEC) and the iso-efficiency map method, the values of efficiency ($\eta_{d\_G\_warm}$)
and MU ($MU_{\eta d\_G\_warm}$) were analysed (Figure 18). This analysis indicates that the iso-efficiency map values ($\eta_{d\_G\_warm\_iso}$) range
from 95.91 % to 81.03 %, while the corresponding IEC efficiency values ($\eta_{d\_G\_IEC}$) range from 96.01 % to 94.71 %. The iso
values generally demonstrate slightly lower efficiency compared to the IEC values. With regard to the MU, the values for the





iso-efficiency map method, $MU_{\eta d\_G\_warm\_iso}$, range from 0.026 % to 0.11 %, while the MU values for the measurements done

to IEC standards, $MU_{\eta d\_G\_IEC}$, range from 0.025 % to 0.038 %. It can be observed that the MU is generally higher with the iso-efficiency map method than when using IEC standards. This analysis provides valuable insights into the comparison between the iso-efficiency map method and IEC standards in terms of efficiency determination and MU for the given operating point profiles. In particular, the differences in efficiency are partly due to the influence of temperature during the measurements. The iso-efficiency map takes longer to measure, providing a longer time period for temperature changes.

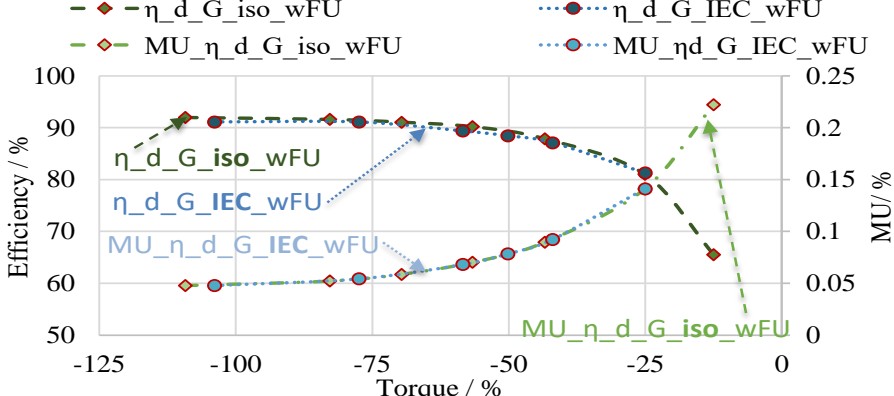

**Figure 18: Direct efficiency with MU for the iso-efficiency map method (iso_wFU) and IEC standards (IEC_wFU) only at nominal torque and nominal rotational speed in warm generator mode with feed to the grid (with frequency converter).**

## 6 Validated iso-efficiency map method used in 4 MW NTB

The successfully validated iso method utilized in the SSTB also proved effective in determining the overall efficiency with relative expanded uncertainty for a 2.75 MW DUT on the 4 MW NTB at RWTH Aachen University. The results are presented

in detail in (Song et al., 2023). As shown in Figure 19, the efficiency (a) including the relative expanded MU (b) was determined traceably at various operating points for the 2.75 MW wind turbine drive train on the 4 MW NTB using the newly developed iso-efficiency map method.

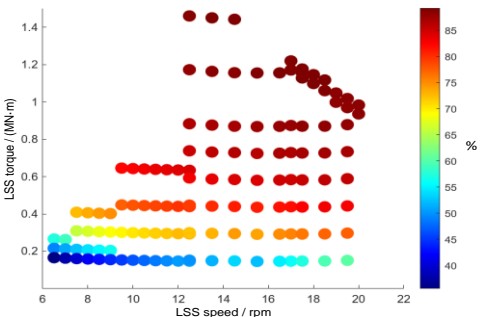

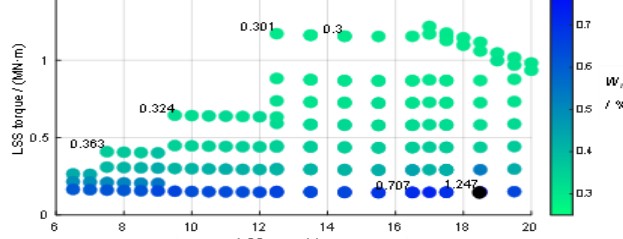

(a) Iso-efficiency map for overall system efficiency.

(b) Overall relative expanded MU for system efficiency of the 2.75 MW FVA research wind turbine drive train.

**Figure 19: Overall efficiency (a) with relative expanded MU (b) for the 4 MW NTB at CWD Aachen** (Song et al., 2023)**.**





## 7 Conclusion and remarks

Tackling climate change is one of the most pressing challenges of our time. In our quest to achieve universal access to affordable energy by 2030 and reach net-zero emissions by 2050 ("Net Zero by 2050 – Analysis - IEA," n.d.), the transition to clean energy, including wind turbines, is playing a pivotal role. In this context, the new iso-efficiency map method proposed in this paper offers promising prospects for testing and validating the total efficiency of DUTs on SSTBs and NTBs. The iso-efficiency map method is a valuable tool for wind turbine testing and analysis of the possible influencing factors. By leveraging

these insights, researchers and practitioners can advance their knowledge and achieve remarkable results in pursuing clean energy goals, particularly by installing energy-efficient wind turbines to complement renewable energy sources.

For the adoption of energy-efficient rotating electrical machines in wind turbines and other applications, effective testing and validation of their total efficiency poses significant challenges. Moreover, determining the total efficiency of wind turbine drive trains in NTBs is a highly complex undertaking due to the intricate nature of these systems. In this paper, a method for

the accurate efficiency determination on NTBs of energy-efficient electrical machines with variable speed drives, as used in wind turbines, was presented and validated. An evaluation of the MU results was also conducted. The iso-efficiency map method can be utilised to map the efficiency of wind turbines during the development process, allowing for further optimisation of system components and the accurate prediction of electrical power generation over their operational lifespan.

The current work described the validation of a direct efficiency determination method, including MU evaluation. The

validation was based on the six-load point profile stipulated in IEC 60034-2-1 and the newly developed iso-efficiency map method with the main focus on the directly determined efficiency of the DUT, in this case an ASM at 400 V and 50 Hz mains supply in generator mode with frequency converter. Both of the load profiles used show good agreement. However, the iso-efficiency map offers further advantage by providing efficiency information over a wide range of torques and rotational speeds. Temperature in particular has an effect on efficiency measurements and the associated MU values in both the SSTB and the

NTB. The study illustrates temperature-related effects such as increased resistance and losses of DUT performance. The temperature has a direct impact on these losses, which are the main factors affecting the overall efficiency of the machine and the entire system. It was shown that the efficiency of the DUTs on both the SSTB and the NTB is higher in the cold state than in the warm state. While an efficiency evaluation is possible in the SSTB for both cold and warm DUTs, in the NTB a disproportionately time-consuming operation is necessary to evaluate changes in efficiency due to temperature. In real wind

turbine operation, there are always temperature fluctuations due to changes in wind speed and in general environmental conditions. Therefore, a standardised method for evaluating the efficiency of wind turbines should include not only torque and rotational speed, but also standardised temperature measurement in order to define operating points and their respective efficiencies.

To make the torque measurement more precise, the zero offset was not only determined statically in one operation position

but averaged over several positions across the entire circumference of the drive train. In contrast to static torque calibration, measurements under rotation require a signal averaging over several revolutions of the drive train in order to adequately take





into account the effects of the sensor's and drive train's own weight as well as additional lateral forces and bending moments due to installation misalignments.

Direct efficiency determination with MU values based on static data from CCs performed on an SSTB compliant with both
IEC standards and the iso-efficiency map method show strong agreement when determining the efficiency and the MU values at rated rotational speed, thus validating the iso-efficiency map method based on IEC standards. However, for a comprehensive assessment of the overall MU, it is imperative to consider the influence of the operating point instability of both electrical and mechanical parameters. This influence of dynamic disturbances on the electrical power measurement, which subsequently affects the efficiency determination with MU values, can be further investigated by analysing mechanical, electrical, and, in
turn, efficiency signal ripples and oscillations.

Using the iso-efficiency map method for direct efficiency determination including MU estimation generally yields higher values, which, however, account for the influence of operating point instability, thereby bringing this evaluation approach more in line with practical reality. Various factors play a role when measuring under rotation. In particular, dynamic effects appear that manifest themselves as torque fluctuations, which again influence the electrical parameters. These influencing
factors become apparent when a high sampling rate is used. They are caused by machine excitation and by design details of the machine that give rise to deviations from ideal sinusoidal magnetic fields and variations in the air gap. The combined effect of torque and rotational speed fluctuations in conjunction with the ripples in current and voltage measurements can be considered by calculating the operating point instability. The broader impacts of this operating point instability need to be taken into account when measuring efficiency, and this will influence both the overall efficiency and the MU estimate.

The DUT's efficiency and associated MU values are inextricably linked to the temperature distribution during operation and to the stability of the mechanical and electrical signal measurements. Even if stationary operation with maximum constancy of the DUT temperature is not always feasible, it is strongly recommended to carry out efficiency measurements in an almost-stationary state with a warmed-up DUT. The precise measurement of mechanical and electrical quantities plays a crucial role in facilitating this endeavour and holds the potential for establishing a traceable efficiency determination methodology for the
test bench. This undertaking aligns with the essential requirements for wind turbines, which in turn contribute to the realisation of the green energy transition towards net-zero emissions in the foreseeable future.

**Competing interests.** The contact author has declared that none of the authors has any competing interests.

**Acknowledgements.** The project 19ENG08 WindEFCY has received funding from the EMPIR programme which is co-financed by the Participating States from the European Union's Horizon 2020 research and innovation programme. The inputs
of all the project partners are gratefully acknowledged.

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
