# Peer review of "Validation of a traceable efficiency determination method for wind turbines with a focus on measurement uncertainty"

_Wind Energy Science, 2023_

## Referee Comment (RC1)

**General comments**

- The contribution explains the application of the iso-efficiency map method for determining the efficiency of rotating electrical machines. The method uses measurements at several combinations of rotation speed and torque for efficiency determination, which goes beyond other methods from existing standards. Measurements are undertaken at a small scale test bench (SSTB) for different operating conditions and measurement setups. It is shown that the iso-efficiency method yields comparable results to standardized methods but provides more insight due to the larger number of load steps considered. Furthermore, an analysis of the MU contribution to the measurements reveals the large importance of operating point instability for efficiency determination. The paper also briefly mentions the application of the method at a 4MW wind turbine nacelle test bench.

- The paper is well structured and allows for fluent reading and good understanding of its contents. The topic and motivation are introduced in appropriate detail and the state of the art is given adequate space and explanation. The paper is written using very well understandable language and grammar.

- The scientific approach and the methods used in the paper are valid and sufficient information to understand the individual steps is given.

- The originality of the paper is good. While it is obvious that the iso-efficiency map method provides more insights compared to standardized procedures since it utilizes more load points, the detailed explanation of the topic, the analysis of the MU and its contributions and the assessment of the temperature influence on MU are a relevant addition to the existing literature on the research topic.

- Due to the focus on measurements from the SSTB, it appears as if the application and demonstration of the method at a wind turbine nacelle test bench is not the focus of the paper. Hence, it appears questionable why the application of the method at the NTB is presented in a dedicated section (section 6), which does not have any subsections and contains only 5 lines of text and a figure taken from another publication. While it is important to point out the relevance of the rather theoretical (scaled-down) study at the SSTB to be utilized in large-scale wind turbine applications, the current form of doing so seems unfavorable, since in the current form section 6 does not provide a significant added value to the overall paper. The authors might either extend section 6 in order to explain similarities and/or differences found at the large-scale application compared to the SSTB, of present the NTB application it in a less prominent form, i.e. not in a dedicated section with only 5 lines of text.

- Formatting of the paper is generally satisfactory, but leaves room for improvement:
  - Font size and style changes in figure captions when including a citation (e.g. Table 6, Figure 6).
  - Formatting of the figures could be improved: some figures are printed too small, so that it is difficult or impossible to read properly if printed at A4. It is apparent that the figure and plot style changes during the different chapters (e.g. font type and size, boxes around plots, see e.g. Fig 1 vs. Fig 3 vs. Fig 7, or Fig 13 vs. Fig 14). This might be harmonized by the authors.
  - The general formatting of figures and mathematical symbols is not ideal and could be improved: some symbols are depicted using underscores instead of subscripts (e.g. lines 298f)

- Terminology "efficiency": The paper elaborates on a method to determine the efficiency of a rotating electrical machine with a possible application in wind energy engineering for drive train / power train efficiency determination. The paper lacks a comprehensive and scientific usage of the term "wind turbine efficiency", since there

is no definition given. In modern wind turbines, the drive train or power train efficiency might be an important factor, but the aerodynamic efficiency, i.e. the ability of converting the wind to rotor torque, is by far the more relevant factor when it comes to "overall wind turbine efficiency" (overall wind turbine efficiency = P_el/P_in = electric power / kinematic power of the wind). By not distinguishing between "overall wind turbine efficiency" and "drive train or power train efficiency" (i.e. mechanical rotor power at the main shaft to electric power output), the importance of the proposed method for drive train efficiency determination might be considerably overvalued when reading the introduction. The authors should distinguish clearly between the different efficiencies considered and comment on the relative importance of drive train efficiency as one factor of overall efficiency.

- Terminology "validation": The title of the paper puts a focus on the validation of the iso-efficiency map method. However, the word "validation" or "validate" is scarcely used throughout the paper, especially in the relevant chapter 5.4. It should be pointed out with more emphasis that the comparison of efficiencies from the different methods leads to a validation of the iso-efficiency map method. It should further be elaborated whether this validation shows a rather high or rather low level of accordance and what this means for further applications.

- The validation approach of the iso-efficiency map method appears to be simple, so that validation should be obtained in almost any case, since the new and the standardized methods take into account the very same load steps for efficiency determination. If the additional load steps of the iso-efficiency map method are not taken into account for validation of the method, how would there be any other differences in the determined efficiency than the ones coming from measurement uncertainties? This is not a general problem of the work, but the simplicity of this validation approach and its implications (e.g. it is very likely to achieve validation) should be elaborated on in this paper.

- A critical error has been made in equation (4) (see specific comments). This error has the potential to propagate throughout the paper since it might be used in (all) later calculations for the different MU terms. It was not possible for the reviewer to check the actual calculations with numbers, so the authors are strongly asked to check for errors originating from equation (4) in their calculations.

**Specific comments**

Introduction

- As stated in the general comments, the usage of the term "wind turbine efficiency" should be more precise. Coming from the importance of high energy conversion efficiency it should be stated that drive train / power train efficiency is only part of this. In general, the function and importance of the wind turbine drive train / power train should be included in the introduction in some more words.

- Page 1, line 28f: it is unclear, how methods traceable to national standards could ensure highest possible efficiency. Efficiency determination methods have no direct effect on the efficiency itself, so they do not lead to higher efficiency.

Section 2

- Page 4, line 76: the Guide to the Expression of Uncertainty in Measurement GUM should be referenced.

- Page 4, line 88: the factor k has not been introduced to this point, which makes it difficult to understand here.

- Equation 4 is wrong, at least if using the given equation 2 as an input. It should be: $d\,eta\,/\,d\,P\_elec = 1\,/\,P\_mech$

d eta / d P_mech = - P_elec / P_mech^2
(all terms with subscript _G)

**Section 3**

- Page 5, line 111: "This approach was chosen because …" – it is not clear to the reviewer what this means and why this is a logical consequence.
- Page 5, line 121f: "torque is applied in descending order", but Figure 3 shows ascending torque steps. Maybe something is wrong here?

**Section 4**

- Page 8, line 185: as stated in general comments, the type and purpose of the "validation" is unclear and not precisely defined. What is validation at all, why do the authors want to validate the method? What is the purpose of the validation, what is an acceptance criterion for the validation to be carried out? There should be more explanation and definition.
- Table 2: the last column is offset
- Table 5: it is unclear what the purpose and unit of the given values is. Is this Nm? Or some relative value (unitless)?
- Table 6: rotor diameter and swept area are not relevant or even applicable properties of a wind turbine nacelle (as stated in the caption).
- Figure 6: the boxes "test bench" and "DUT" appear to be offset

**Section 5**

- Figure 7: The torque level numbers given in the plot appear to be offset and are difficult to match with the plot.
- Figure 7: Torque is given with symbol "M", despite using "T" for torque in the rest of the paper.
- Figure 8: font is hard to read in A4
- Figure 9: torque "M" vs "T"
- Figure 10: for better comparability between (a) and (c) as well as (b) and (d) the authors might use the same scale. Font sizes are too small.
- The introduction to section 5.2 is not fully understandable to the reviewer (lines 305-310:
  - What do you mean by "It also shows the losses when the number of operating points decrease"? – Figure 11 does not show any losses.
  - Line 307f: How does interpolation facilitate to "improve the evaluation process"? Interpolation is independent of the new iso-efficiency map method.
  - Line 309f: figure 11 does not show efficiency, therefore it can not "offer guidance for informed decisions regarding the DUT's efficiency".
- Line 317: in contrast to the "warm state" (thermally stable), the "cold state" could require some more introduction and/or definition. By running a test, the "cold state" would not be at a constant temperature due to heating up during execution of the test runs. Do all "cold state" measurements start at the same reference temperature? What if tests last for different durations (i.e. the end of the test at "cold state" has a higher or lower temperature compared to another one in "cold state")? This might be elaborated in more words.
- Figure 12: The caption states that "measured mechanical parameters of torque and rotational speed" are included, but they are not visible in the figure.
- Page 16, line 335: How does "measuring under thermal steady state conditions" (directly) allow to "assess the effects of temperature-related factors" – since it is only one measurement, no influence of temperature could be derived?

- Figure 13 and Figure 14 show the same or similar data as before but in a different format. It should be explained in the text or in the figure caption how this matrix-style depiction (fig 13) or characteristic map style (fig14) has been derived from the measurements undertaken.
- Page 19, line 375f: figure 16 does not show "each component's contribution from equation 10", but just the overall sum.
- Line 377: it is unclear to the reviewer, what the "low values of the torque measurement (about 10%)" refer to.
- Figure 18: It should be highlighted that is the key figure to validation of the iso-efficiency map method.

Section 6

- As stated in general comments, it is unclear why such a short section 6 is a benefit to the paper (please see suggestions above).

Section 7

- Page 23, line 448f: The iso-efficiency method is applied to wind turbine drive trains / power trains, not to "wind turbine testing and [the] analysis of possible influencing factors". It is still a very important contribution, but the application should be described more precisely here.
- Page 23, line 457: it is unclear, how the iso-efficiency map method would be applied during the development process of a wind turbine, since it requires (prototype) testing of the system.

**Technical corrections**

- The paper shows several occurrences of double parentheses in text expressions, which should be avoided (e.g. p2, line 45).
- Page 2, line 38f: The source for the EMPIR project WindEFCY might be chosen in a more elegant way to avoid the n.d. (not dated) mention in the reference.
- Page 3, line 58: references to IEC 60034-2-1 and IEC 60034-2-3 are given in different styles, both of which leave room for improvement (identical wording in text and ref for the first, see also line 61; wrong ref style for the latter)
- Page 4, line 74: units are irrelevant should not be given in the text since equations 1 and 2 are dimensionless.
- Equation 3: both terms in the root should have equal sizing of the square brackets.
- Equation 3 uses subscripts for terms (e.g. eta_d_G), while eq. 4 uses a version with underscores. This applies to several occurrences in the paper and should be harmonized.
- P5 line 110: (Song et al. 2023) is not the proper reference to show what the iso-efficiency map method is (see two lines above).
- P6, line 140: "test benches for wind turbines" might include rotor blade tests, etc. Be more precise, e.g. "NTBs".
- P15, line 298f: use proper symbol formatting with subscripts instead of underscores.
- Figure 11: the caption state 45 measurement points, but there are only 43 as stated in the other text.
- Line 326: instead of referencing Fig12a it should be Fig13a.
- Line 340: the highest efficiency of 91.968% can not directly be seen from figure 14. Why does this value have 3 fractional digits, while all others have 2?
- Figure 15 is hard to read

- Line 421: Figure 9 does not show a difference of 4.9°K, but rather of 5.3°K (107.9 – 102.6)
- Line 421: Figure 9 does not have subfigure a)
- Line 423: Figure 12 does not have subfigure a)
- Figure 18: style of the symbols is not uniform and should be improved in this figure
- Line 438: "iso-efficiency map method"